# LncRNA *JHDM1D-AS1* Is a Key Biomarker for Progression and Modulation of Gemcitabine Sensitivity in Bladder Cancer Cells

**DOI:** 10.3390/molecules28052412

**Published:** 2023-03-06

**Authors:** Isadora Oliveira Ansaloni Pereira, Glenda Nicioli da Silva, Tamires Cunha Almeida, Ana Paula Braga Lima, André Luiz Ventura Sávio, Katia Ramos Moreira Leite, Daisy Maria Fávero Salvadori

**Affiliations:** 1Departamento de Análises Clínicas, Pharmacy School, UFOP—Federal University of Ouro Preto, Ouro Preto 35400-000, MG, Brazil; 2Laboratory of Pain and Signaling, Butantan Institute, Sao Paulo 05503-900, SP, Brazil; 3Departamento de Odontologia, Faculdade do Centro Oeste Paulista—FACOP, Piratininga 17490-000, SP, Brazil; 4Departamento de Ciências Médicas, Universidade do Oeste Paulista—UNOESTE, Jaú 19050-900, SP, Brazil; 5Departamento de Cirurgia, Medical School, USP—University of São Paulo, São Paulo 05508-060, SP, Brazil; 6Departamento de Patologia, Medical School, UNESP—São Paulo State University, Botucatu 01151-000, SP, Brazil

**Keywords:** bladder cancer, *JHDM1D-AS1*, long non-coding RNAs

## Abstract

Long non-coding RNAs are frequently found to be dysregulated and are linked to carcinogenesis, aggressiveness, and chemoresistance in a variety of tumors. As expression levels of the *JHDM1D* gene and lncRNA *JHDM1D-AS1* are altered in bladder tumors, we sought to use their combined expression to distinguish between low-and high-grade bladder tumors by RTq-PCR. In addition, we evaluated the functional role of *JHDM1D-AS1* and its association with the modulation of gemcitabine sensitivity in high-grade bladder-tumor cells. J82 and UM-UC-3 cells were treated with siRNA-*JHDM1D-AS1* and/or three concentrations of gemcitabine (0.39, 0.78, and 1.56 µM), and then submitted to cytotoxicity testing (XTT), clonogenic survival, cell cycle progression, cell morphology, and cell migration assays. When *JHDM1D* and *JHDM1D-AS1* expression levels were used in combination, our findings indicated favorable prognostic value. Furthermore, the combined treatment resulted in greater cytotoxicity, a decrease in clone formation, G0/G1 cell cycle arrest, morphological alterations, and a reduction in cell migration capacity in both lineages compared to the treatments alone. Thus, silencing of *JHDM1D-AS1* reduced the growth and proliferation of high-grade bladder-tumor cells and increased their sensitivity to gemcitabine treatment. In addition, the expression of *JHDM1D/JHDM1D-AS1* indicated potential prognostic value in the progression of bladder tumors.

## 1. Introduction

According to the Global Cancer Observatory, bladder cancer accounted for 212,536 cancer-related deaths worldwide in 2020. It is approximately four times more prevalent in men than in women and is the 6th most prevalent cancer in men worldwide [1]. Urothelial tumors (tumors on the urothelial cells, specialized transitional bladder cells) account for 90% of bladder cancer subtypes, of which 75% are non-muscle-invasive and 25% are muscle-invasive, attacking the bladder’s musculature and adjacent structures. The classification of high- or low-grade for tumors refers to the degree of cell differentiation and is an important prognostic factor. Non-muscle-invasive bladder cancers (NMIBC) are typically superficial, low-grade tumors with favorable prognoses but high recurrence rates; their treatment consists primarily of transurethral resection and intravesical therapies. In contrast, muscle-invasive tumors (MIBC) are high-grade neoplasias that are associated with worse prognosis, especially when evolving metastasis to pelvic lymph nodes [2,3,4,5].

The most common therapies used for neoadjuvant chemotherapy are MVAC (methotrexate, vinblastine, doxorubicin, and cisplatin) and CG (cisplatin and gemcitabine). The CG regimen is related to a higher pathological complete response and a better toxicity profile when compared to the MVAC regimen [6,7]. Gemcitabine is a prodrug that is similar to the nucleoside deoxycytidine and is used in CG protocols. Its active form, gemcitabine triphosphate, exerts an antitumor effect by weakly inhibiting DNA polymerase, depleting deoxyribonucleotide reserves, and interrupting DNA strands by its repair-resistant incorporation. The toxic effects of gemcitabine include myelosuppression and hepatotoxicity [8].

Long non-coding RNAs (lncRNAs) show highly specific expression patterns in several tissues and biological contexts, revealing their regulatory roles in physiological processes (such as embryogenesis and cell differentiation) and pathological processes. LncRNAs can regulate gene expression via chromatin interactions and epigenetic regulation, but also at transcriptional and post-transcriptional levels, interacting with genes, other RNAs, or specific proteins [9,10,11]. Some lncRNAs are often found dysregulated in several tumors, and an increasing number of studies have demonstrated their roles as oncogenic or tumor suppressors [12]; lncRNAs are also related to chemoresistance processes [13]. The abnormal expression of over 2000 lncRNAs in bladder tumors demonstrates a clear association with tumor progression, invasiveness, and drug resistance. Therefore, lncRNAs can be investigated as potential biomarkers, prognostic indicators, and therapeutic targets in bladder cancer [14]. The bladder cancer-specific lncRNA *UCA1* (urothelial cancer associated 1) was also associated with tumor progression, invasiveness, and chemoresistance in bladder cancer cell lines, by competing with tumor-suppressing miRNAs and modulating signaling pathways [15,16,17]. Overexpression of the lncRNA *MALAT-1* was also observed in bladder cancer cells, which was related to increased cell migration and invasion, and chemoresistance, by activation of the Wnt pathway and subsequent promotion of epithelial–mesenchymal transition, and by sponging miR-125b [18,19]. Overexpressed *MALAT-1* was also associated with higher-grade and metastatic bladder tumors, making it a good diagnostic and prognostic biomarker for these cases [20].

Long non-coding RNAs can also take part in carcinogenesis by regulating the angiogenesis processes that are mediated by inflammatory mechanisms. The histone demethylase *JHDM1D* (also known as *KDM7A*) is an epigenetic regulator found to be evolved in growth, drug resistance, and progression in several tumors [21,22,23]. The lncRNA *JHDM1D-AS1*, an antisense transcript of the *JHDM1D* gene (Appendix A), is also overexpressed in tumor cells and tissues under nutritional stress. This transcript is related to tumor growth by stimulating angiogenesis via the activation of pro-inflammatory and pro-angiogenic factors. Increased expression of *JHDM1D-AS1* has been found in gastric and lung cancer cells [24,25]. The platform “The Atlas of Non-Coding RNAs in Cancer (TANRIC)” revealed the expression of the lncRNA *JHDM1D-AS1* in bladder cancer cell lines, including J82 and UM-UC-3, and showed higher expression of this lncRNA in higher-grade tumoral tissues than in low-grade tumors. In addition, data found on the TANRIC platform indicate that higher expression of *JHDM1D-AS1* is associated with a lower survival probability in patients with bladder cancer [26].

Thus, this study aimed to elucidate the functional role of the lncRNA *JHDM1D-AS1* and its association with the modulation of gemcitabine sensitivity in J82 and UM-UC-3 high-grade bladder-tumor cells. Furthermore, we investigated whether gene/lncRNA expression in urothelial carcinoma tissues can be used to distinguish between low- and high-grade tumors. We intend to search for novel therapeutic strategies and biomarkers for the classification and prognosis of bladder cancer.

## 2. Results

### 2.1. JHDM1D-AS1 and JHDM1D Expression in Bladder Tumor Samples

Despite the higher expression of lncRNA *JHDM1D-AS1* in high-grade tumor samples, no significant difference was observed (Figure 1a). Nevertheless, *JHDM1D* gene expression was 2.01 times greater in high-grade tumors than in low-grade tumors (*p* = 0.0392) (Figure 1b). The analysis of the correlation between *JHDM1D* and *JHDM1D-AS1* expression revealed a moderately positive correlation (r = 0.7204, *p* < 0.0001) (Figure 1c). In addition, analysis of ROC curves showed that the combination of *JHDM1D* and *JHDM1D-AS1* had potential prognostic value (*p* = 0.028), demonstrating their ability to distinguish between low- and high-grade tumors (Figure 1d).

### 2.2. Cell Viability and Morphology

Compared to untreated and siRNA-*JHDM1D-AS1* treated cells, J82 cells treated with 1.56 µM of gemcitabine exhibited a significant reduction in viability. In addition, in cells treated with siRNA-*JHDM1D-AS1* combined with 0.39 and 0.78 µM gemcitabine, there was a significant reduction compared to the respective gemcitabine concentration alone (Figure 2a). Compared to untreated cells, cell viability decreased significantly in all treated UM-UC-3 cells (Figure 2b).

Figure 3 shows that treatments with siRNA-*JHDM1D-AS1* alone or in combination with gemcitabine significantly reduced the cell density of J82 and UM-UC-3 lines compared to untreated cells. Furthermore, the treatments induced alterations in cell morphology, including long cell extensions, irregular appearance, rounding, and death.

### 2.3. Clonogenic Survival

All treatments significantly decreased the clonogenic survival of J82 cells relative to untreated cells. Compared to cells treated with siRNA-*JHDM1D-AS1*, clonogenic survival was significantly reduced in this cell line after treatment with gemcitabine at 0.78 and 1.56 µM and after treatment with all gemcitabine and siRNA-*JHDM1D-AS1* combinations. In addition, the combinations of gemcitabine 0.39 and 1.56 µM with siRNA-*JHDM1D-AS1* caused a significant reduction in the cell colony formation compared to the respective concentration of gemcitabine alone (Figure 4a).

In UM-UC-3 cells, all treatments led to a significant decrease in clonogenic survival compared to untreated cells. In all combination treatments, clonogenic survival was reduced compared to siRNA-*JHDM1D-AS1* alone and the respective concentration of gemcitabine alone (Figure 4b).

### 2.4. Cell Cycle Progression

In the J82 cell cycle analysis, cell cycle arrest was observed at the G0/G1 phase in the cells treated with all concentrations of gemcitabine and with gemcitabine and siRNA-*JHDM1D-*AS1. Cell cycle arrest at the G0/G1 phase was significantly higher in the groups submitted to the combined treatment with gemcitabine and siRNA-*JHDM1D-AS1*, when compared to the groups treated solely with the respective gemcitabine concentrations. This interference was accompanied by a diminished population of cells in S and G2/M phases. In addition, there was an increase in the sub-G1 content in the cells treated with gemcitabine and siRNA-*JHDM1D-AS1* alone (Figure 5 and Appendix A).

Similarly, in the UM-UC-3 cell line, there was mild cell cycle arrest at the G0/G1 phase in the cells treated with all concentrations of gemcitabine and with gemcitabine and siRNA *JHDM1D-AS1*. The combined treatment led to a significant increase in cell populations in the S phase, when compared to cells treated solely with the respective gemcitabine concentrations and to the untreated control. Additionally, an increase in the sub-G1 content was observed in the cells treated with gemcitabine alone. In the UM-UC-3 cells treated with siRNA *JHDM1D-AS1* alone, there was cell cycle arrest at the S phase (Figure 6 and Appendix A).

### 2.5. Cell Migration

In J82 cells, we could observe that all treatments reduced the cell migration compared to untreated cells, in both analyzed times. In 24 h, there was also a significant reduction in the migration process after the treatment with gemcitabine at 0.39 and 0.78 µM and with the two highest combination treatments when compared to siRNA-*JHDM1D-AS1* alone. The association of gemcitabine 1.56 µM and *siRNA-JHDM1D-AS1* also reduced the J82 cell migration significantly compared with the same concentration of gemcitabine alone (Figure 7a,b). After 48 h of treatment, there was a consistent reduction in cell migration compared to untreated cells, for all treatments. In addition, all the combined treatments reduced cell migration significantly when compared to the groups treated solely with gemcitabine (Figure 7a,c).

For UM-UC-3 cells, a reduction in the migration process was observed after all 24 h treatments compared to the untreated cells (Figure 8a,b). After 48 h, this effect was maintained in the cells treated with siRNA-*JHDM1D-AS1*, with gemcitabine at 1.56 µM, and with all combinations of gemcitabine and siRNA-*JHDM1D-AS1*. In addition, there was a significant reduction in the number of migrated cells after treatment with 0.39 or 0.78 µM of gemcitabine combined with siRNA-*JHDM1D-AS1* compared with the respective concentration of the chemotherapy compound alone (Figure 8a,c). Importantly, although there was no inhibition of migration after treatment with gemcitabine, it is possible to observe a reduction in cell density, which is consistent with the results of cell viability (Appendix A).

## 3. Discussion

Non-coding long RNAs play an important role in carcinogenesis [24,27]. Therefore, a better understanding of the functional role of lncRNAs could help in cancer diagnosis, prognosis, and treatment. Based on this, first, we conducted a marker lesion study using *JHDM1D* and lncRNA *JHDM1D-AS1* expression to distinguish between low-and high-grade bladder tumors. The results showed different expression levels of *JHDM1D* in low- and high-grade tumors, suggesting a possible role of this gene in bladder tumor progression. This gene regulates many biological processes, including differentiation, development, and the growth of several cancer cells [23,28]. *JHDM1D* was found to be upregulated in prostate cancer tissue, and its chemical inhibition reduced proliferation and induced apoptosis of prostate cancer cells [22]. Nevertheless, Osawa et al. [29] demonstrated that this gene is overexpressed in tumor cells under nutritional stress and is associated with the suppression of tumor growth via down-regulation of pro-angiogenic factors in cancer cells and xenograft mouse models. This discrepancy may indicate a different role of *JHDM1D* in different tumor conditions and stages [23].

Antisense lncRNAs are transcribed from the promoter regions of the coding gene and can affect not only the expression of sense genes but also the expression of distant genes [30]. Similarly to Kondo et al. [31], we also identified a long noncoding antisense transcript, *JHDM1D-AS1*, whose expression increased proportionally to *JHDM1D* levels. The expression *JHDM1D-AS1* did not differ between low- and high-grade samples, even though higher expression levels of the referred lncRNA are observed in high-grade tumors. The number of tissue samples analyzed may have been a limitation of the present study. A greater number of bladder tumor specimens could enable a clearer demonstration of the different expression levels of lncRNA *JHDM1D-AS1*. Indeed, data available from The Cancer Genome Atlas (TCGA), comprising a total of 252 tumors and 19 normal samples, show higher expression of this lncRNA in high-grade tumors compared to low-grade tumors. However, when *JHDM1D* and *JHDM1D-AS1* expression levels were used in combination, potential prognostic value for the progression of bladder tumors was indicated. Thus, *JHDM1D*-AS1 silencing in high-grade cell lines, subsequently resulting in changes in biological behavior, should be considered.

Nowadays, gemcitabine is present in several standard protocols for muscle-invasive bladder cancer chemotherapy due to its efficacy and better toxicity profile [32]. Chemotherapy resistance is a significant barrier to cancer treatment, as it drastically reduces the efficacy of treatment and is strongly associated with tumor progression and recurrence. Several studies have demonstrated the influences of lncRNAs in the modulation of cellular pathways involved in chemoresistance, including in bladder cancer [13,19,33,34,35]. Despite the increased levels of lncRNA *JHDM1D-AS1* in bladder cancer cells, there were no studies investigating the functional relevance of this lncRNA in high-grade bladder cancer cells or the connection of this lncRNA to gemcitabine treatment in these tumors. Consequently, in the second part of this study, we examined the functional role of *JHDM1D-AS1* and the effects of combining various gemcitabine concentrations with *JHDM1D-AS1* inhibition on high-grade bladder-tumor cell lines and their chemosensitivity.

Initially, the effect of silencing *JHDM1D-AS1* expression was examined. Cytotoxicity related to the knockdown of this lncRNA in bladder carcinoma cells was observed. Some authors have observed that this lncRNA can be targeted to inhibit other tumor types. Wu et al. suggested that lncRNA JHDM1D-AS1 promote gastric cancer progression by upregulating oncogenic PRAF2 level by trapping miR-450a-2-3p [24]. Moreover, Yao et al. showed that, in small cell lung cancer, JHDM1D-AS1 binding attenuates proteasome-mediated degradation of DHX15, an ATP-dependent RNA helicase involved in ribosome biogenesis, enhancing growth and metastasis [25]. In comparison to the isolated chemical, it was observed that gemcitabine combined with siRNA-*JHDM1D-AS1* decreased the viability of J82 cells, but not of UM-UC-3 cells. Additionally, the clonogenic survival assay was performed to establish the long-term effects of the treatment. Evaluation of the clone formation is one of the most important tests for determining the effect of a chemotherapy drug on tumor cells, as it indicates if the cells lost the ability to divide and proliferate [36]. In the two investigated cell lines, the combinations of gemcitabine and the siRNA for *JHDM1D-AS1* decreased the proliferative capacity of the cells relative to the drug alone and the treatment with the siRNA alone. These findings indicate that silencing *JHDM1D-AS1* improved the sensitivity of tumor cells to chemotherapy. It should be noted that, in both cells, treatment with siRNA alone or in combination led to a reduction in colony formation that was greater than the XTT-observed reduction in cell viability. These results also indicate that the treatment’s impact is mediated by DNA-damaging processes that comprise the reproductive integrity of cells. In particular, Da Silva et al. [37] showed that, in bladder cancer cell lines, gemcitabine induces cytostatic effects that, according to the findings of the present investigation, seem to be more apparent following lncRNA suppression. Consistently with previous findings, morphological changes indicative of loss of adhesion and cell death, such as cells with irregular and rounded shapes and the presence of cell debris, were more pronounced after the combined treatment of gemcitabine and siRNA-*JHDM1D-AS1*, along with a decrease in cell density, compared to treatments with the isolated compounds.

Cell migration plays an essential role in tumor invasion and metastasis, and can be assessed by observing cell migration, after treatment, through a gap in the cell monolayer [38]. The relationship between long non-coding RNAs and tumor cell metastasis has already been reported by Dhamija et al. [39]. In addition to modulating the regulation of genes related to metastasis, these molecules contribute to the regulation of the epithelial–mesenchymal transition (EMT) process and act on in vitro cell invasiveness and migration [39]. Furthermore, lncRNA *JHDM1D-AS1* has been linked with the development and metastasis of non-small cell lung cancer and the proliferation and migration of gastric cancer cells [24,25]. In the present study, *JHDM1D-AS1* silencing reduced cell migration in both cell lines when compared to the untreated control. Moreover, gemcitabine concentrations combined with *JHDM1D-AS1* silencing led to a decrease in cell migration when compared to treatment with gemcitabine alone or with siRNA in both cell types, and the effects become more pronounced after 48 h. Thus, it appears that the lncRNA *JHDM1D-AS1* is associated with the metastatic process and enhances the anti-migratory effects of gemcitabine in high-grade bladder-carcinoma cells.

Studies on the impact of gemcitabine on the cell cycle have yielded apparently inconsistent results about cell cycle kinetics [37,40]. G1/G0-phase arrest was detected in the two *TP53-*mutant cell lines treated with gemcitabine, and it was slightly but significantly increased in J82 cells when gemcitabine was combined with *JHDM1D-AS1* silencing. Despite a slight increase in UM-UC-3 cell population in the G1/G0 phase, the most pronounced cell cycle perturbations detected following the combined treatment were cell cycle arrests in the S and G2/M phases. Thus, the more pronounced alterations in cell cycle kinetics were observed after treatment with gemcitabine and lncRNA inhibition.

In conclusion, our findings indicate that the long non-coding RNA *JHDM1D-AS1* may be related to high-grade bladder cancers. In J82 and UM-UC-3 high-grade bladder-cancer cell lines, the silencing of this lncRNA produced decreased in cell viability and clone formation, morphological alterations, and decreased cell migration. In addition, the silencing of *JHDM1D-AS1* combined with gemcitabine treatment resulted in enhanced cytotoxicity, higher rates of cell death, lower cell migration capacity, and cytostatic effects when compared to treatment with the isolated compound. Therefore, it appears that *JHDM1D-AS1* plays an important role in the progression of bladder cancer, and its silencing increases the gemcitabine sensitivity of high-grade tumor cells. In future investigations, inhibiting this lncRNA may be considered as a strategy for achieving a therapeutic response with a lower concentration of gemcitabine, hence decreasing side effects.

## 4. Materials and Methods

### 4.1. Clinical Samples

A total of 30 fresh bladder cancer tissue samples, 20 histologically diagnosed as high-grade tumors and 10 as low-grade tumors, were collected at the University of Sao Paulo Biorepository (Sao Paulo, Brazil) and Amaral Carvalho Hospital’s tumor repository (Jau, Sao Paulo, Brazil). All tumor samples were collected via transurethral resection and histopathologically classified by a pathologist. The grading and stage were determined according to the World Health Organization (WHO) systems and Tumor-Node-Metastasis (TNM) 2017. All patients were male.

The study was approved by the Ethics Committee of the Sao Paulo State University (protocol 48193715.6.0000.5411), and all methods were performed following the approved guidelines.

### 4.2. Cell Lines, Culture Conditions, and Reagents

Two urothelial bladder cancer cell lines, acquired from the Cell Bank of Rio de Janeiro, Brazil, were used in the in vitro experiments: i. J82, J82, derived from high-grade (grade III) tumor with three point mutations in the *TP53* gene (two in exon 8 (codon 271: Glu (GAG) → Lys (AAG); codon 274 Val (GTT) → Phe (TTT)) and one in exon 9 (codon 320 Lys (AAG) → Asn (AAC)) [41,42,43]; ii. UM-UC-3, derived from a high-grade tumor with a point mutation at TP53 (exon 4, codon 113: Phe (TTG) → Cys (TCG)) [41,42,43].

Both cell lines were cultivated in monolayers, using DMEM culture medium supplemented with 10% bovine serum and 100 U/mL of penicillin G, 100 U/mL of streptomycin (Sigma-Aldrich, Saint Louis, EUA), and 2.5 µg/mL of amphotericin B (Cristália, Itapira, Brazil), and maintained at 37 °C in an atmosphere of 5% CO_2_. All experiments were performed with exponentially growing cells [37]. Gemcitabine (Gemzar) was obtained from Eli Lilly do Brazil Ltd.a. Treatment with gemcitabine was set to 24 h. Dilutions were performed in nuclease-free water.

### 4.3. Expression and Knockdown Assays

Tissue biopsies were snap-frozen and stored at −80 °C. Total RNA was isolated using the RNeasy Mini Kit^®^ (Qiagen, Hilden, Germany) according to the manufacturer’s protocol. RNA concentration and purity were determined using a NanoDrop spectrophotometer (Thermo Scientific, Waltham, MA, USA). RNA quality was analyzed using a 2100 Bioanalyzer (Agilent, Santa Clara, CA, USA), and only samples with an RNA integrity number (RIN) ≥ 6.0 were used. Complementary DNA (cDNA) was synthesized using the High Capacity Kit (Applied Biosystems, Waltham, MA, USA) with random priming according to the manufacturer’s instructions. Expression levels of the *JHDM1D/KDM7A* gene and *JHDM1D-AS1* lncRNAs were analyzed using RT-qPCR. Endogenous reference genes (*HSPCB* and *ACTB*) were selected using the NormFinder software, version 5 [44].

For lncRNA *JHDM1D-AS1* knockdown, J82 and UM-UC-3 cells were transfected with SMARTpool Lincode Human *JHDM1D-AS1* siRNA (Horizon Discovery Ltd., Cambridge, UK), specific to human *JHDM1D-AS1*. Lipofectamine RNAiMAX (Invitrogen Life Technologies) was used as transfection agent for the siRNA-*JHDM1D-AS1*, according to the manufacturer’s protocol. Cells transfected with lincode non-targeting control siRNAs were used as a negative control. *JHDM1D-AS1* knockdown in the two cell lines was confirmed 72 h post-transfection, by RT-qPCR (Appendix A). Total RNA was extracted with Quick-RNA™ MicroPrep ZymoSpin™ IC Columns kit and dosed in NanoDrop^®^. cDNA was confectioned using High Capacity^®^ kit (Applied Biosystems), according to the manufacturer’s protocol. RTq-PCR was performed using SYBR Green (Thermo Fisher Scientific). *GAPDH* gene was used as endogenous control for normalization. Treatment time with siRNA-*JHDM1D-AS1* was set to 48 h.

### 4.4. Cytotoxicity Assay (XTT)

The XTT assay (Cell Proliferation Kit II, ROCHE Diagnostics, Mannheim, Germany) was first used to define three concentrations of gemcitabine to be used in later experiments with J82 and UM-UC-3 cells. Gemcitabine concentrations were defined as 0.39, 0.78, and 1.56 µM. To evaluate the cytotoxic effects after treatment with gemcitabine + si-*JHDM1D-AS1,* 1 × 10^4^ cells were seeded into 24-well culture plates, and 24 h later, treated with siRNA-*JHDM1D-AS1* (10 pmol). Forty-eight hours later, cells were treated with three different concentrations of gemcitabine (0.39 µM, 0.78 µM, and 1.56 µM) for 24 h. Untreated cells; cells treated with non-targeting siRNA, used as controls; and cells treated only with si-*JHDM1D-AS1* or only with gemcitabine, were cultured in parallel. After treatment, cells were washed with Hank’s solution (0.4 g KCl, 0.06 g KH_2_PO_4_, 0.04 g Na_2_HPO_4_, 0.35 g NaHCO_3_, 1g glucose, and 8 g NaCl in 1000 mL H_2_O) and incubated at 37 °C with tetrazolium salt XTT (ROCHE Diagnostics, Mannheim, Germany) for 5 h. Immediately after incubation, absorbance was read in a spectrophotometer at a wavelength of 450 nm. Experiments were performed in technical triplicate.

### 4.5. Clonogenic Survival Assay

In order to evaluate the long-term effects of the treatment on J82 and UM-UC-3 cells, a clonogenic survival assay was performed. First, 3 × 10⁵ cells were seeded into 12-well culture plates and incubated for 24 h at 37 °C in an atmosphere of 5% CO_2_. After 24 h, cells were treated with siRNA-*JHDM1D-AS1* (10 pmol) for 48 h and then treated with 0.39, 0.78, or 1.56 µM of gemcitabine for 24 h. Untreated cells and cells treated with non-targeting siRNA, used as controls, along with cells treated only with si-*JHDM1D-AS1* or only with gemcitabine, were cultured in parallel. Past treatment time, the cells were washed with Hank’s solution, trypsinized, re-plated (1 × 10^3^ cells) into 12-well culture plates, and incubated at 37 °C in an atmosphere of 5% CO_2_. After 10 days (when untreated controls reach maximum confluence for colony formation experiences), the culture medium was removed, and cells were washed with Hank’s solution, fixed with 4% formaldehyde solution for 20 min, hydrated with 100% methanol for 20 min, and stained with crystal 0.5% violet solution dissolved in 25% methanol. A 33% acetic acid solution was used to remove the dying solution, and the contents of the plates were transferred to a 96-well plate. Absorbance was measured in a spectrophotometer at a wavelength of 570 nm [45] and was used to assess the percentage of colonies formed, that is, the reproductive capacity of the cells. Experiments were performed in technical triplicate.

### 4.6. Cell Morphology

To observe the effects of gemcitabine + si-*JHDM1D-AS1* treatment on the morphologies of J82 and UM-UC-3 cells, 2 × 10^5^ cells of each cell line were seeded into 12-well culture plates, incubated for 24 h, and treated as described above. After treatment, the cells were observed and photographed under a phase-contrast optical microscope, at 200× magnification [46]. Experiments were performed in technical triplicate.

### 4.7. Wound Healing Assay

The wound healing assay was used to appraise cell migration after the combined treatment of gemcitabine and lncRNA *JHDM1D-AS1* knockdown. For that, 4 × 10⁵ J82 and UM-UC-3 cells were seeded into 12-well culture plates and incubated at 37 °C and 5% of CO_2_; for 24 h. After incubation time, cells were treated with siRNA-*JHDM1D-AS1* (10 pmol) for 48 h and then treated with 0.39, 0.78, or 1.56 µM of gemcitabine for 24 h. Immediately after gemcitabine treatment, a smooth and linear scratch was made in the middle of the cell monolayer, using a 200 µL pipette tip. Untreated cells and cells treated with non-targeting siRNA were used as controls, as cells treated only with siRNA-*JHDM1D-AS1* or only gemcitabine were cultured in parallel. Cell migration over the scratch area was observed 24 and 48 h after gemcitabine treatment, at 40× magnification in an optical microscope with inverted light. Cell migration was evaluated and quantified using ImageJ^®^ software (Adapted from Lima et al., 2020 [45]). Experiments were performed in technical triplicate.

### 4.8. Cell Cycle Progression

To analyze cell cycle alterations after the treatment with siRNA-*JHDM1D-AS1* and gemcitabine, 4 × 10^5^ J82 and UM-UC-3 cells were seeded into 12-well culture plates for 24 h. Then, cells were treated with siRNA-*JHDM1D-AS1* (10 pmol) for 48 h and afterward treated with 0.39, 0.78, or 1.56 µM of gemcitabine for 24 h. After treatment time, cells were washed with Hank’s solution, trypsinized, and centrifuged at 1000 rpm for 10 min. The supernatant was discarded, and the sediment was fixed with 70% ethanol and maintained at −20 °C for at least 12 h. Then, cells were washed with Hank’s solution, centrifuged, resuspended with 200 μL of labeling solution containing propidium iodide, and maintained in ice, protected from light, for 30 min [47]. Cell cycle kinetics were obtained by flow cytometry (BD FACSCalibur), and 20,000 events were detected. The percentages of cells in the Sub G1, G0/G1, S, and G2/M phases were analyzed using FlowJo^®^ software. Experiments were performed in technical triplicate.

### 4.9. Statistical Analysis

The nonparametric Mann–Whitney test was used for differential gene expression analyses in the clinical samples, and the values are expressed as the mean ± SD. The relationships between the differentiated values were examined using Spearman’s rank correlation test (r = correlation coefficient), according to Akoglu (2018) [48]. A receiver operating characteristic (ROC) curve was constructed, and the area under the curve (AUC) was calculated to assess the specificity and sensitivity of the *JHDM1D* gene/lncRNA *JHDM1D-AS1* in differentiating high- and low-grade tumors. Statistical significance was set to *p* < 0.05. Data from the cytotoxicity, clonogenic, cell cycle, and migration assays were analyzed using one-way ANOVA followed by Tukey’s multiple comparison test. Statistical significance was set to *p* < 0.05. All statistical analyses were performed using GraphPad Prism^®^ 6.

## Figures and Tables

**Figure 1 molecules-28-02412-f001:**
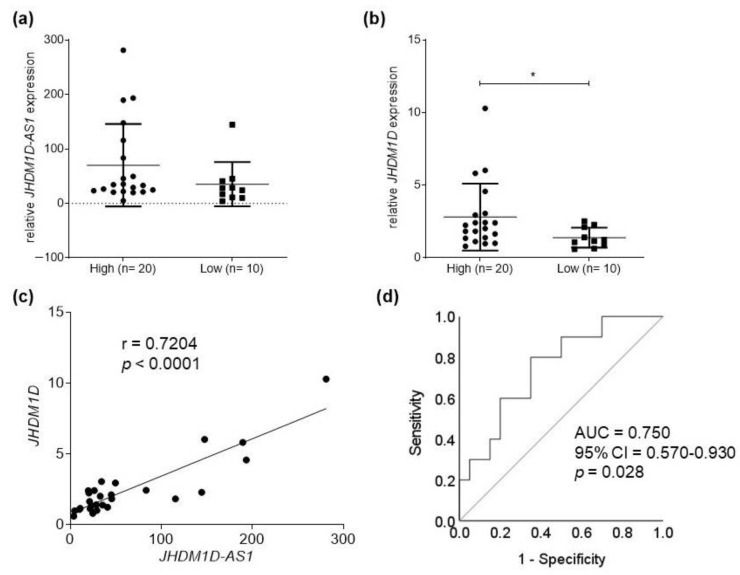
Expression of the *JHDM1D* gene (**a**) and *JHDM1D-AS1* lncRNA (**b**) in patients with low- and high-grade bladder tumors. (**c**) Moderately positive correlation between *JHDM1D* and *JHDM1D-AS1* expression (r = 0.7204) in low- and high-grade bladder tumor samples (Spearman correlation analysis). (**d**) Receiver operating characteristics (ROC) curves using the combination between *JHDM1-AS1* and *JHDM1D* to distinguish between low- and high-grade bladder tumors. * *p* < 0.05.

**Figure 2 molecules-28-02412-f002:**
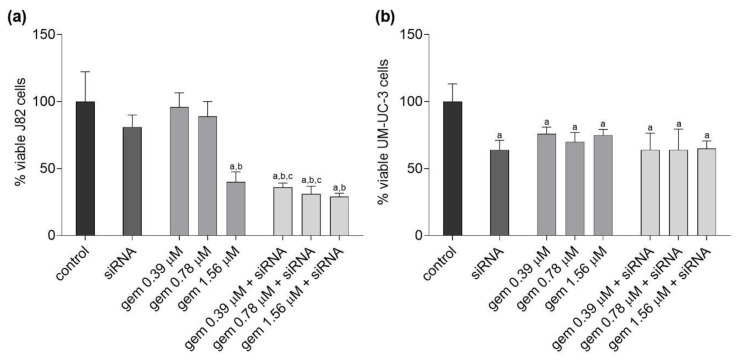
Percentage of viability in J82 (**a**) and UM-UC-3 (**b**) cell lines after treatment with gemcitabine, siRNA-*JHDM1D-AS1*, or gemcitabine combined with siRNA-*JHDM1D-AS1*. Control: untreated cells; gem: gemcitabine. a: *p* < 0.05 compared to control; b: *p* < 0.05 compared to siRNA; c: *p* < 0.05 compared to respective gemcitabine concentration alone.

**Figure 3 molecules-28-02412-f003:**
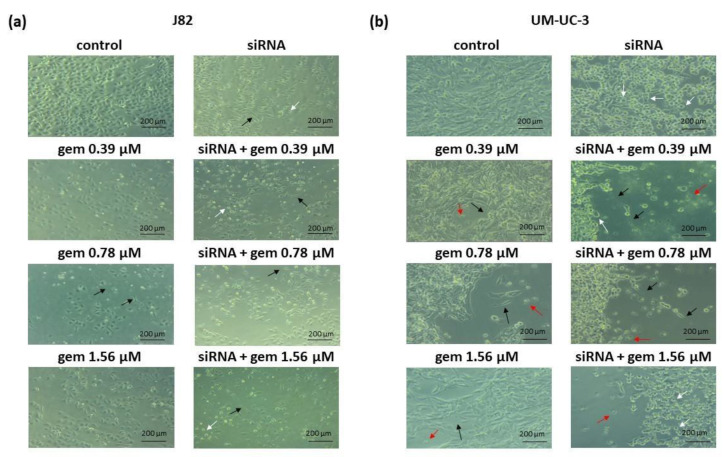
Morphologies of J82 **(a)** and UM-UC-3 (**b**) cells after treatment with gemcitabine, siRNA-*JHDM1D-AS1,* or gemcitabine combined with siRNA-*JHDM1D-AS1*. Control: untreated cells; gem: gemcitabine. Black arrows: elongated cells; white arrows: round cells; red arrows: dead cells. Phase-contrast microscope, ×200.

**Figure 4 molecules-28-02412-f004:**
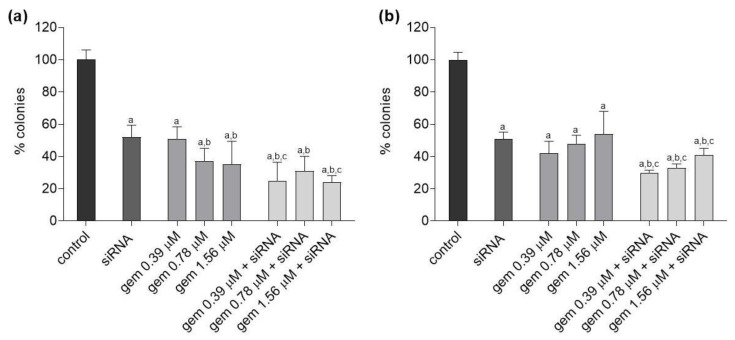
Percentages of J82 (**a**) and UM-UC-3 (**b**) colonies after treatment with gemcitabine, siRNA *JHDM1D-AS1*, or gemcitabine combined with siRNA *JHDM1D-AS1*. Control: untreated cells; gem: gemcitabine. a: *p* < 0.05 compared to control; b: *p* < 0.05 compared to siRNA; c: *p* < 0.05 compared to respective gemcitabine concentration alone.

**Figure 5 molecules-28-02412-f005:**
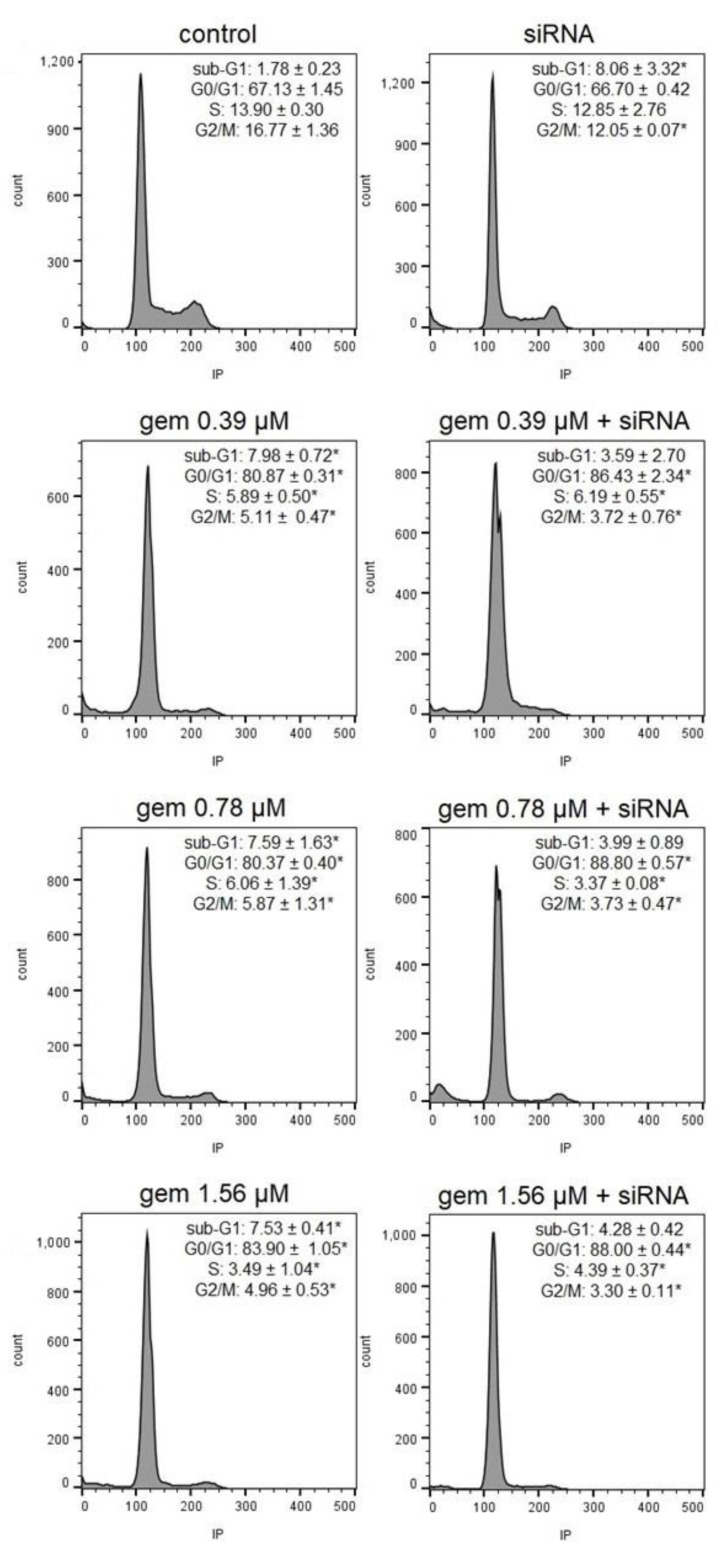
Representative histograms and percentages of J82 cells in each phase after treatment with gemcitabine, siRNA *JHDM1D-AS1*, or gemcitabine combined with siRNA *JHDM1D-AS1*. Control: untreated cells; gem: gemcitabine; IP: propidium iodide. * *p* < 0.05 compared to control.

**Figure 6 molecules-28-02412-f006:**
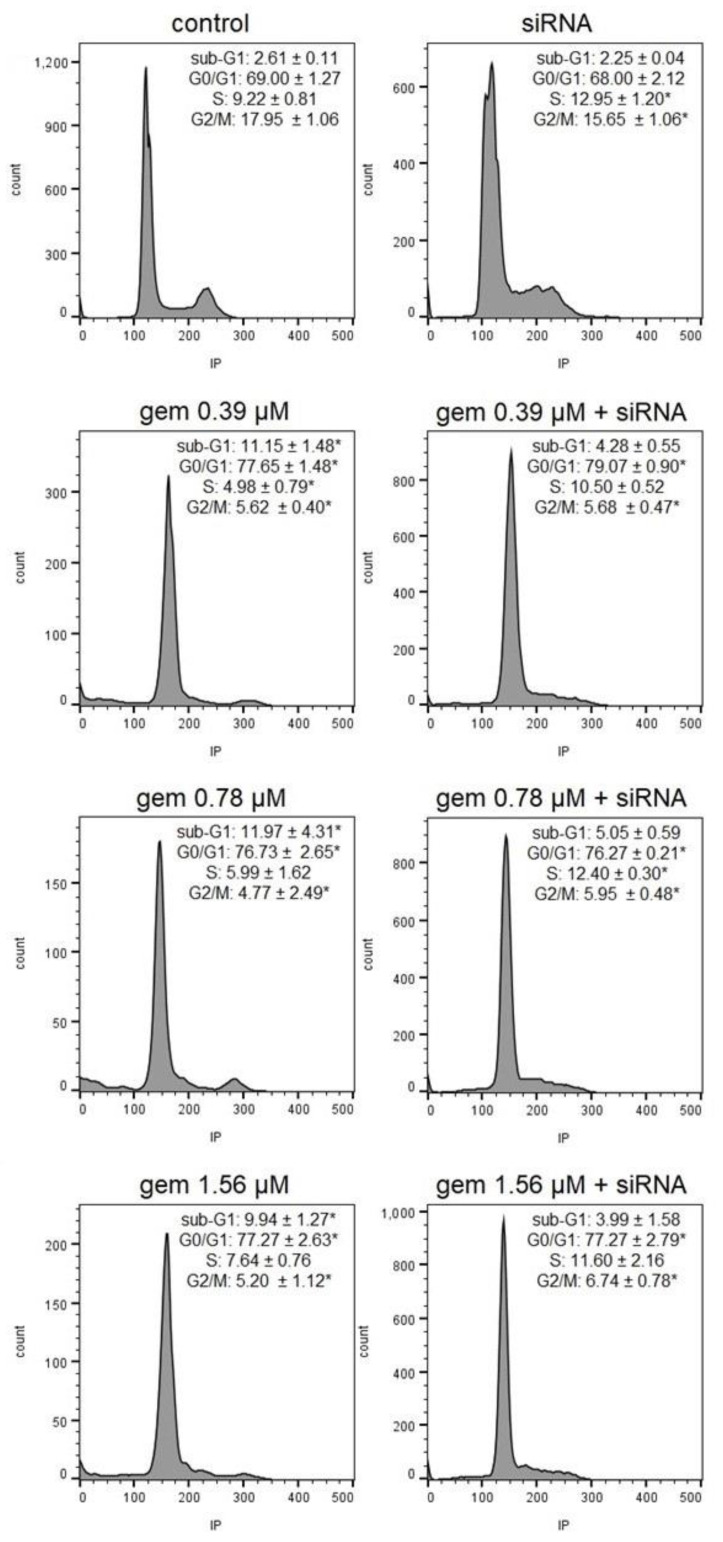
Representative histograms and percentages of UM-UC-3 cells in each phase after treatment with gemcitabine, siRNA *JHDM1D-AS1*, or gemcitabine combined with siRNA *JHDM1D-AS1*. Control: untreated cells; gem: gemcitabine; IP: propidium iodide. * *p* < 0.05 compared to control.

**Figure 7 molecules-28-02412-f007:**
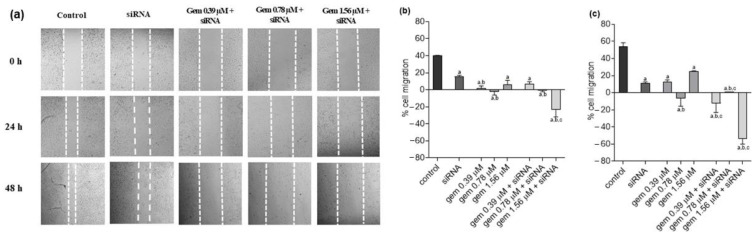
Photographs (**a**) and quantification (**b**,**c**) of cell migration in J82 cells after treatment with gemcitabine, siRNA *JHDM1D-AS1*, or gemcitabine combined with siRNA *JHDM1D-AS1*. Control: untreated cells; gem: gemcitabine. a: *p* < 0.05 compared to control; b: *p* < 0.05 compared to siRNA; c: *p* < 0.05 compared to respective gemcitabine concentration alone.

**Figure 8 molecules-28-02412-f008:**
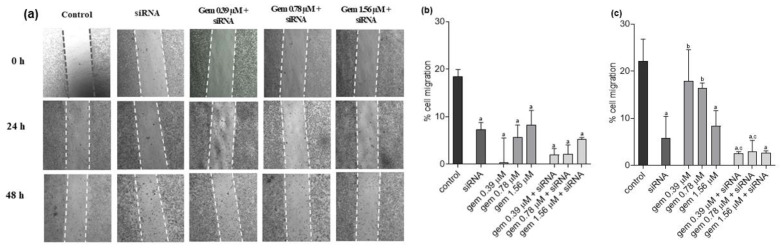
Photographs (**a**) and quantification (**b**,**c**) of cell migration in UM-UC-3 cells after treatment with gemcitabine, siRNA *JHDM1D-AS1*, or gemcitabine combined with siRNA *JHDM1D-AS1*. Control: untreated cells; gem: gemcitabine. a: *p* < 0.05 compared to control; b: *p* < 0.05 compared to siRNA; c: *p* < 0.05 compared to respective gemcitabine concentration alone.

## Data Availability

Not applicable.

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
