# Peer review of "LncRNA JHDM1D-AS1 Is a Key Biomarker for Progression and Modulation of Gemcitabine Sensitivity in Bladder Cancer Cells"

_molecules, 2023, doi:10.3390/molecules28052412_

Round 1
Reviewer 1 Report
Pereira et al. demonstrated that silencing of JHDM1D-AS1 inhibit tumor cell growth and proliferation, also increased their sensitivity to gemcitabine treatment. This is an interesting study but lacks some mechanistic validation.
The authors successfully demonstrated that inhibiting JHDM1D-AS1 expression using siRNA could reduce tumor cell proliferation. Using two cell lines is sufficient to validate these findings. However, there are several major concerns need to address:
1. Supplementary Figure 1 showed the quality control of the siRNA treatment, but how the expression level of JHDM1D itself was regulated through the siRNA treatment was not shown.
2. In UM-UC-3 cell line experiments (Figure 2, 3 and 4), it seems siRNA treatment itself could decrease tumor growth and viability, however, adding gem with different concentrations did not increase the inhibition rate or affect viability at all. Increasing sensitivity to gemcitabine seems to work only in J82 cells. If this is the case, the authors should try to understand what’s the mechanism behind it. For example, an RNA-seq focusing on those two cell lines with different treatment would work. The authors should at least provide possible insights into why this treatment of siRNA targeting JHDM1D-AS1 could work, and what’s the potential mechanism behind those, in this situation.
3. The authors tested their hypothesis in their own cohort of bladder cancer patients. TCGA also has a large cohort of bladder cancer patients. The authors should use TCGA and other available datasets to further verify the correlation of JHDM1D-AS1 and patient’s prognosis.
Reviewer 2 Report
In this study, the authors identified the JHDM1D-AS1 as a biomarker in bladder cancer cells. The manuscript is well-written and the results are well-organized. My suggestions/comments are here:
- The structural organization of a JHDM1D and lncRNA JHDM1D-AS1 gene (with length) should be provided in figure 1.
- The expression level of reference genes (HSPCB, ACTB, etc.) could be provided in (figure 1a, 1b. Also, include the information on a number of samples/data points.
- It seems that in Figure 1a, the median expression of lncRNA JHDM1D-AS1 is closer to 80 (in high) and around 40 in low-grade cancer cells. However, the statistical test showed this difference is not significant. It requires more explanation.
- Figure 3 is interesting! I wonder if the authors have done any RNA sequencing to explain this phenotype in more detail.
- The resolution/quality of Figure 5 should be improved
Round 2
Reviewer 1 Report
The authors addressed most of my concerns